# Multi-Condition Remaining Useful Life Prediction Based on Mixture of Encoders

**DOI:** 10.3390/e27010079

**Published:** 2025-01-17

**Authors:** Yang Liu, Bihe Xu, Yangli-ao Geng

**Affiliations:** 1Key Laboratory of Big Data & Artificial Intelligence in Transportation (Ministry of Education), Beijing Jiaotong University, Beijing 100044, China; fourcm@hotmail.com (Y.L.); 22120445@bjtu.edu.cn (B.X.); 2China Energy Railway Equipment Co., Ltd., Beijing 100011, China

**Keywords:** RUL prediction, deep learning, mixture of encoders, transformer

## Abstract

Accurate Remaining Useful Life (RUL) prediction is vital for effective prognostics in and the health management of industrial equipment, particularly under varying operational conditions. Existing approaches to multi-condition RUL prediction often treat each working condition independently, failing to effectively exploit cross-condition knowledge. To address this limitation, this paper introduces MoEFormer, a novel framework that combines a Mixture of Encoders (MoE) with a Transformer-based architecture to achieve precise multi-condition RUL prediction. The core innovation lies in the MoE architecture, where each encoder is designed to specialize in feature extraction for a specific operational condition. These features are then dynamically integrated through a gated mixture module, enabling the model to effectively leverage cross-condition knowledge. A Transformer layer is subsequently employed to capture temporal dependencies within the input sequence, followed by a fully connected layer to produce the final prediction. Additionally, we provide a theoretical performance guarantee for MoEFormer by deriving a lower bound for its error rate. Extensive experiments on the widely used C-MAPSS dataset demonstrate that MoEFormer outperforms several state-of-the-art methods for multi-condition RUL prediction.

## 1. Introduction

The Remaining Useful Life (RUL) prediction of machinery and components is a cornerstone of prognostics and health management, playing a pivotal role in ensuring the safety, reliability, and efficiency of industrial systems. RUL refers to the estimated time period during which a system or component is expected to perform its intended function before failure occurs. Accurate RUL predictions facilitate optimized maintenance schedules, reduce downtime, and minimize costs associated with unexpected failures. With the rapid advancements of Industry 4.0, RUL prediction has gained increasing importance across sectors such as aerospace, energy, manufacturing, and transportation, where system performance and reliability are critical [1,2]. However, the complexity of modern industrial systems, especially under dynamic operating conditions, presents significant challenges for accurately estimating RUL. To address these challenges, researchers have developed a wide range of RUL prediction methods, which can broadly be categorized into physics-based models, traditional machine learning approaches, and, more recently, deep learning techniques.

Physics-based models rely on first-principles knowledge, such as material degradation processes, structural dynamics, and failure mechanisms, to estimate RUL. These models are valued for their interpretability and the ability to incorporate domain expertise into them. However, their application is often limited by the need for precise system modeling and difficulties in capturing complex or nonlinear degradation patterns [3]. In contrast, traditional machine learning approaches, such as support vector machines, random forests, and regression models, have been widely employed for RUL prediction due to their capability to uncover patterns in historical sensor data and model nonlinear degradation processes [4]. While these methods have demonstrated promising results, they often require extensive feature engineering and face challenges in handling high-dimensional or time-series data, limiting their scalability and generalization performance.

In recent years, deep learning techniques have emerged as powerful tools for RUL prediction owing to their ability to automatically extract high-level features from large-scale sensor data. Architectures such as convolutional neural networks (CNNs) and recurrent neural networks (RNNs) have shown remarkable improvements in RUL prediction accuracy [5]. Additionally, attention-based models, including Transformers, and hybrid frameworks that combine multiple neural architectures have further expanded the horizons of RUL prediction [6,7]. The flexibility and robustness of deep learning make it particularly well suited for addressing the challenges posed by varying operating conditions and environments. However, many real-world systems operate under multi-condition scenarios, where environmental and operational variability substantially influence degradation behavior. Effectively modeling and generalizing across such scenarios remains a key challenge in RUL prediction [8,9]. Existing models often struggle to fully exploit the intricate relationships across operating conditions, leaving ample room for further innovation.

To address these limitations, this paper introduces MoEFormer, a novel framework specifically designed for multi-condition RUL prediction. MoEFormer integrates a Mixture of Encoders (MoE) architecture with a Transformer-based framework to enhance feature extraction, cross-condition knowledge fusion, and temporal dependency modeling. In contrast to existing methods, which often approach multi-condition RUL prediction in isolation, MoEFormer employs specialized encoders tailored to capture degradation features under specific operating conditions. A gated mixture module dynamically fuses these condition-specific features, enabling the model to leverage shared knowledge while preserving condition-specific insights. Additionally, the incorporation of Transformer layers improves the model’s ability to capture long-range temporal dependencies, which are crucial for accurate RUL estimation.

The main contributions of this paper are as follows:We propose a novel multi-condition RUL prediction framework that dynamically integrates condition-specific and shared knowledge using a Mixture-of-Encoders- and Transformer-based architecture.We derive a theoretical lower bound for the error rate of the proposed MoEFormer, providing a formal performance guarantee.Through extensive experiments on the C-MAPSS dataset, we demonstrate that MoEFormer outperforms existing state-of-the-art methods in predictive accuracy and robustness.

## 2. Related Work

### 2.1. Deep Learning for Sequence Modeling

The Remaining Useful Life (RUL) prediction problem can be abstractly formulated as a sequence modeling task. Deep learning has emerged as a powerful paradigm for sequence modeling, offering significant advantages over traditional statistical methods in capturing complex temporal dependencies. A major milestone in this domain was the development of recurrent neural networks (RNNs) [10] and their variants, such as long short-term memory (LSTM) networks [11] and gated recurrent units (GRUs) [12]. These architectures addressed the vanishing gradient problem in standard RNNs, enabling the modeling of long-range dependencies in sequential data. Convolutional neural networks (CNNs), though traditionally associated with image processing, have also been adapted for sequence modeling. Temporal convolutional networks [13] utilize causal convolutions and dilations to capture long-term dependencies in time-series data, often demonstrating superior computational efficiency and training stability compared to RNN-based methods.

The advent of attention mechanisms has further revolutionized sequence modeling by enabling models to selectively focus on relevant parts of input sequences regardless of their length. The Transformer architecture, introduced by Vaswani et al. [14], replaced recurrence with self-attention mechanisms and positional encodings, setting new benchmarks in sequence modeling. Transformer derivatives, such as BERT [15] and GPT [16], have inspired extensive exploration of attention-based architectures for time-series data. In this study, we leverage the Transformer architecture as the foundational model for sequence modeling.

### 2.2. Deep Learning for RUL Prediction

The application of deep learning to RUL prediction has garnered significant interest due to the nonlinear and complex nature of degradation processes in industrial systems. RNNs have been instrumental in modeling time-series data for RUL prediction. Early work by Malhi et al. [17] employed Elman RNNs to model degradation stages, laying the foundation for sequence-based architectures in predictive maintenance. The introduction of LSTM significantly enhanced the ability to capture long-term dependencies. For instance, Yuan et al. [18] and Wu et al. [19] trained LSTM models to directly predict RUL, while Zheng et al. [20] improved performance by integrating feed-forward layers on top of LSTM networks. Ellefsen et al. [21] incorporated unsupervised pretraining with restricted Boltzmann machines to enhance feature extraction alongside genetic algorithms for hyperparameter optimization. More recently, Wang et al. [5] introduced the ISSA-LSTM, utilizing an improved sparrow search algorithm for optimized LSTM parameter configuration. Similarly, Pent et al. [22] proposed a dual-channel LSTM network that adaptively extracts temporal features and their first-order derivatives for RUL prediction. Deng et al. [23] proposed a novel hybrid MC-GRU-PF model, achieving real-time degradation monitoring and RUL prediction of the ball screw where the analytical measurement model is not available.

Beyond RNN-based methods, CNNs have also shown strong promise in RUL prediction. CNNs excel at feature extraction through convolution and pooling operations, making them well suited for multi-channel sensor data. Babu et al. [6] and Li et al. [24] adopted CNNs to automatically extract temporal features, followed by feed-forward neural networks for RUL regression. Xu et al. [25] extended this approach by incorporating dilated convolutions, which enabled broader temporal context capture and faster training.

Hybrid architectures that integrate multiple deep learning models have further demonstrated the potential of deep learning in RUL prediction. For example, Liu et al. [26] combined LSTM networks with Gaussian process regression to provide both accurate predictions and uncertainty quantification. Preprocessing techniques, such as empirical mode decomposition, were employed to extract meaningful sensor patterns, especially in battery health monitoring. Zhu et al. [27] integrated CNNs and RNNs to fuse spatial and temporal features, achieving strong performance in RUL prediction. Similarly, Xu et al. [28] proposed a multi-source information fusion framework, combining CNNs and LSTMs to extract frequency- and time-domain features. Deng et al. [29] proposed a calibration-based hybrid transfer learning framework to improve the data fidelity and model generality of the hybrid physical and data-driven prognosis model for RUL prediction of rolling bearings across different machines. Jia et al. [30] presented a joint distribution adaptation-based transfer network with diverse feature aggregation, where the diverse feature aggregation module enhances feature extraction capability across large domain gaps and the joint maximum mean discrepancy is adopted to reduce the distribution discrepancy automatically. Ref. [31] combined the theoretical bound and deep adversarial network for machinery open-set diagnosis transfer, aiming to improve the diagnosis performance in open-set scenarios where some classes in the target domain are not seen during training.

Attention mechanisms have further enriched RUL prediction by enabling models to focus on critical features and time steps. Sun et al. [32] introduced an attention-based framework that fused handcrafted and learned features, improving prediction accuracy. Chen et al. [7] applied attention mechanisms to bearing RUL prediction, dynamically refining feature selection during training. Recent advances, such as the feature-sequence attention module by Cen et al. [33] and the feature reuse multi-scale attention residual network by Song et al. [34], have significantly advanced the accuracy of RUL predictions. Deng et al. [35] presented a double-layer attention-based adversarial network for partial transfer learning in machinery fault diagnosis, which can effectively transfer knowledge from the source domain to the target domain and improve the diagnosis accuracy even when there is a distribution discrepancy between the two domains.

In summary, although there are several prior works for addressing multi-condition RUL prediction, such as MODBNE [36], Cap-LSTM [8], and CNN-LSTM [37], these methods either incorporate condition-specific inputs as additional features or rely on domain adaptation techniques to align feature distributions. However, these approaches often fail to explicitly model cross-condition knowledge sharing or specialize in condition-specific feature extraction. In contrast, MoEFormer introduces a novel Mixture of Encoders (MoE) architecture, enabling the dynamic fusion of shared and condition-specific knowledge, and a Transformer-based predictor for capturing long-range temporal dependencies. This architectural innovation allows MoEFormer to outperform these methods, as demonstrated by significant reductions in RMSE and Score metrics on multi-condition datasets.

## 3. Preliminary

### 3.1. Transformer for Sequence Modeling

The Transformer model, introduced by Vaswani et al. [14], has become the cornerstone of many state-of-the-art deep learning architectures for sequence modeling tasks. It utilizes a self-attention mechanism to capture dependencies between elements of the input sequence without relying on the recurrent or convolutional structures traditionally used in sequence modeling. Given an input sequence $X=\{x_{1},x_{2},\ldots ,x_{n}\}$ of length *n*, where each element $x_{i}$ is a vector in $R^{d}$, the Transformer processes this input sequence through multiple layers of self-attention and feed-forward networks to extract useful features. The input sequence is first embedded into a sequence $X_{\mathrm{emb}}\in R^{n\times d}$ with positional encodings to retain information about the relative positions of tokens in the sequence.

The core of the Transformer model is the self-attention mechanism. In this mechanism, each element in the sequence is compared with every other element, allowing the model to weigh the importance of different tokens when generating a representation. The self-attention operation computes a weighted sum of input vectors, where the weights are determined by the similarity between the tokens. Given an input sequence of embeddings $X_{\mathrm{emb}}$, the self-attention mechanism computes the following:(1)$$H=\mathrm{Attention}\left(Q,K,V\right)=\mathrm{softmax}\left(\frac{QK^{T}}{\sqrt{d}}\right)V$$
where *Q*, *K*, and *V* are the query, key, and value matrices, respectively. These are computed from the input embeddings $X_{\mathrm{emb}}$ as follows:(2)$$Q=X_{\mathrm{emb}}W_{Q},\;K=X_{\mathrm{emb}}W_{K},\;V=X_{\mathrm{emb}}W_{V}$$
where $W_{Q}$, $W_{K}$, and $W_{V}$ are learned weight matrices. The softmax function ensures that the attention weights are normalized and sum to one, enabling the model to focus on the most relevant tokens in the sequence.

The attention block is followed by a position-wise feed-forward network, which consists of two fully connected layers with a ReLU activation in between:(3)$$z_{i}=\max\left(0,h_{i}W_{1}+b_{1}\right)W_{2}+b_{2}\;\left(i=1,2,\ldots ,n\right)$$

After passing through the self-attention and feed-forward layers, the final representation of the sequence is obtained.

For sequence modeling tasks where the goal is to output a single feature from the sequence, the Transformer model typically uses a pooling operation or a linear layer applied to the final output. Let $z_{i}\in R^{d}$ represent the output of the final layer for the *n*-th token. The output feature $y\in R^{d}$ can be obtained by applying a pooling operation, such as(4)$$y=\mathrm{Pooling}(z_{1},z_{2},\ldots ,z_{n}).$$

This produces a single output, which can be used for tasks such as regression or classification. The Transformer architecture is highly effective for sequence modeling tasks due to its ability to capture long-range dependencies via the self-attention mechanism, making it a powerful tool for RUL prediction.

### 3.2. RUL Prediction

Remaining Useful Life (RUL) prediction refers to the task of estimating the time duration until a system or component is expected to fail or reach a critical threshold beyond which its performance is no longer acceptable. From a machine learning perspective, RUL prediction is formulated as a regression problem, where the objective is to predict a continuous value representing the remaining time (or cycles) until failure, based on observed sensor data up to the current time. These sensor data provide critical information about the state of the equipment, and the goal of the prediction model is to infer the RUL from these data.

Let $x_{t}\in R^{d}$ represent the feature vector at time *t*, where *d* is the dimensionality of the input feature space. The output of the model is the predicted RUL, denoted as $\hat{r}\in R$, which corresponds to the expected number of time steps or cycles remaining before failure. Formally, the RUL prediction problem can be expressed as(5)$$\hat{r}=f\left(x_{T-L+1},x_{T-L+2},\ldots ,x_{T};\theta \right)$$
where the following variables are used:f(·) is the learned model function;θ represents the model parameters, which are learned during training;$x_{T-L+1},x_{T-L+2},\ldots ,x_{T}$ are the sensor data that describe the system’s state within a window of length *L* ending at time *T*.

In many practical scenarios, the equipment may operate under varying working conditions, which can significantly affect the sensor data distribution. For example, a jet engine may operate at different Mach numbers, throttle resolver angles, altitudes, and ambient temperatures, leading to considerable differences in the sensor readings under each condition. As a result, RUL prediction becomes more challenging when these variations in operating conditions are present, as the data distribution under different conditions may be substantially different.

This paper specifically addresses the RUL prediction task under multiple working conditions. In this context, the sensor datum $x_{t}$ at time step *t* is collected from a specific working condition:(6)$$x_{t}\in C_{k}\;\left(t=T-L+1,\ldots ,T\;\mathrm{and}\;k=1,\ldots ,K\right),$$
where $C_{k}$ represents the *k*-th working condition. Given the sensor data ${\left\{x_{t}\right\}}_{t=T-L+1}^{T}$ from *K* distinct working conditions ${\left\{C_{k}\right\}}_{k=1}^{K}$, the goal is to build a model *f* (as defined in (Equation 5)) that minimizes the prediction error. The error is typically quantified using a loss function, such as the mean squared error (MSE):(7)$$E\left[{\left(\hat{r}-r\right)}^{2}\right],$$
where *r* is the true RUL value. This paper proposes MoEFormer, a Mixture of Encoders (MoE)-based model, to address the challenges of RUL prediction under multiple working conditions.

## 4. Methods

Figure 1 illustrates the framework of MoEFormer. The proposed MoEFormer consists of three primary components: **distribution alignment**, **Mixture of Encoders**, and **Transformer predictor**. The distribution alignment step aims to mitigate distribution drift across sensor features collected under varying operating conditions. The Mixture of Encoders component is designed to extract complementary knowledge from different working conditions, thereby generating a more comprehensive feature encoding for sensors across different times. Finally, the Transformer predictor is employed to capture temporal dependencies within the features, ultimately yielding the final RUL prediction. These components are discussed in detail below.

### 4.1. Distribution Alignment

Sensor feature sequences often originate from different working conditions, each of which may exhibit significantly different statistical properties. The goal of this module is to eliminate these discrepancies and align the feature distributions across varying operating conditions. To achieve this, we first compute the mean and standard deviation for the *k*-th working condition as follows:(8)$$\mu_{k}=\frac{1}{|C_{k}|}\sum_{x_{t}\in C_{k}}x_{t},\;\sigma_{k}=\sqrt{\frac{1}{|C_{k}|}\sum_{x_{t}\in C_{k}}{\left(x_{t}-\mu_{k}\right)}^{2}},$$
where $C_{k}$ denotes the set of sensor features under the *k*-th working condition. For a sensor feature $x_{t}$ at time *t*, we assume it was collected under working condition $C_{c_{t}}$, where $c_{t}\in \left\{1,2,\ldots ,K\right\}$ is the index of the working condition. The feature sequence is then aligned as follows:(9)$${\bar{x}}_{t}=\frac{x_{t}-\mu_{c_{t}}}{\sigma_{c_{t}}}\;\mathrm{for}\;t=T-L+1,T-L+2,\ldots ,T,$$
resulting in the aligned features ${\left\{{\bar{x}}_{t}\right\}}_{t=T-L+1}^{T}$, which facilitate the learning of subsequent model parameters.

### 4.2. Mixture of Encoders

While distribution alignment helps normalize sensor features from different working conditions, it does not imply that sensor characteristics across conditions fully reflect consistent equipment behavior. In fact, sensor data from different conditions contain complementary information about the equipment’s operating status. To leverage this complementary information, we introduce the **Mixture of Encoders** (MoE) module.

This module includes *K* feature encoders, corresponding to the *K* operating conditions. Each encoder specializes in processing the features from the *k*-th working condition. To integrate knowledge from all encoders, a gate control network is introduced to mix the outputs of the different encoders. The process is formalized as follows:(10)$$\begin{array}{cc} e_{tk} & ={\mathrm{Encoder}}_{k}\left({\bar{x}}_{t}\right)\;\left(k=1,\ldots K\right),\\ \alpha_{t} & =\mathrm{gate}({\bar{x}}_{t}),\\ e_{t} & =\sum_{k=1}^{K}\alpha_{tk}e_{tk}, \end{array}$$
where $\alpha_{tk}$ is the weight assigned to the feature extracted by ${\mathrm{Encoder}}_{k}$. The gate network consists of a two-layer fully connected network with ReLU activation, and the output is normalized using a softmax function.

Although each encoder is designed to specialize in the features from its respective working condition, we found that directly training the network end to end does not always yield optimal results. Therefore, we introduce an auxiliary loss to guide each encoder towards focusing on features from the corresponding working condition. Specifically, we use the working condition index $c_{t}$ of $x_{t}$ to apply a cross-entropy loss to the gate network’s output:(11)$$L_{aux}=-\frac{1}{L}\sum_{t=T-L+1}^{T}\sum_{k=1}^{K}1\left\{k=c_{t}\right\}\cdot \log\alpha_{tk},$$
where 1{p} is an indicator function that equals 1 if *p* is true and 0 otherwise.

### 4.3. Transformer Predictor

Once the features are processed through the Mixture of Encoders, we obtain a mixed-feature vector for each time step. While these features effectively represent the equipment’s status at a given time, the RUL prediction requires modeling the temporal dependencies among these features. To this end, MoEFormer introduces a Transformer predictor.

Given a sequence of mixed feature vectors ${\left\{e_{t}\right\}}_{t=T-L+1}^{T}$, we employ the Transformer architecture, as described in Section 3.1, to model the temporal dependencies:(12)$${\left\{z_{t}\right\}}_{t=T-L+1}^{T}=\mathrm{Transformer}\left({\left\{e_{t}\right\}}_{t=T-L+1}^{T}\right).$$

Since the final RUL prediction depends on the entire sequence, we introduce an averaging operation over the sequence:(13)$$z=\frac{1}{L}\sum_{t=T-L+1}^{T}z_{t}.$$

Additionally, as noted by [38], the sensor time series trend contains valuable RUL-related information. To enhance the feature representation, MoEFormer incorporates this information by calculating the Pearson correlation coefficient between the aligned feature sequence ${\left\{{\bar{x}}_{t}\right\}}_{t=T-L+1}^{T}$ and the time indices ${\left\{t\right\}}_{t=T-L+1}^{T}$:(14)$$c=\frac{\sum_{t=T-L+1}^{T}\left({\bar{x}}_{t}-\mathrm{mean}\left({\left\{{\bar{x}}_{t}\right\}}_{t=T-L+1}^{T}\right)\right)\left(t-\mathrm{mean}({\left\{t\right\}}_{t=T-L+1}^{T})\right)}{\sqrt{\sum_{t=T-L+1}^{T}{\left({\bar{x}}_{t}-\mathrm{mean}\left({\left\{{\bar{x}}_{t}\right\}}_{t=T-L+1}^{T}\right)\right)}^{2}\sum_{t=T-L+1}^{T}{\left(t-\mathrm{mean}({\left\{t\right\}}_{t=T-L+1}^{T})\right)}^{2}}},$$
where $c\in R^{d}$ is the trend coefficient vector. This vector is then concatenated with the output *z* from the Transformer and passed through a fully connected layer to produce the final RUL prediction:(15)$$\hat{r}=\mathrm{FC}\left([z,c]\right).$$

The RUL loss is computed using the mean squared error between the predicted $\hat{r}$ and the true RUL *r*:(16)$$L_{RUL}={\left(\hat{r}-r\right)}^{2}.$$

Finally, combining the RUL loss (Equation 16) and the auxiliary loss (Equation 11), we obtain the total training loss:(17)$$L=L_{RUL}+\lambda L_{aux},$$
where λ is a hyperparameter controlling the weight of the auxiliary loss, which is set to λ=0.7 in this study.

## 5. Theoretical Analysis of MoEFormer

In this section, we derive a probabilistic error bound for MoEFormer. Specifically, we analyze the probability of a prediction result, denoted by $\hat{r}$, having an error not exceeding ϵ:(18)$$p^{\left(\epsilon \right)}=P\{|\hat{r}-r\left|\leq \epsilon \right\}.$$

Intuitively, this probability decreases as ϵ decreases. Now, consider the prediction error probability $p_{\mathrm{MoE}}^{\left(\epsilon \right)}$ for MoEFormer, as well as the error probability $p_{\mathrm{Enc}}^{\left(\epsilon \right)}$ for a model utilizing only a single encoder. We aim to demonstrate that the error probability of MoEFormer, $p_{\mathrm{MoE}}^{\left(\epsilon \right)}$, is bounded below by a quantity that depends on $p_{\mathrm{Enc}}^{\left(\epsilon \right)}$. Formally, we present the following theorem:

**Theorem** **1.**
*
**(Probabilistic error bound for MoEFormer).**
*
* Suppose K encoders process features independently, and a trained gate network assigns input features to encoders such that the ratio of the probability of assigning a sample to the correct encoder versus an incorrect encoder is 1+α, where α>0. Then, the prediction error probability of MoEFormer is bounded below by*

(19)
$$p_{\mathrm{MoE}}^{\left(\epsilon \right)}\geq \frac{\left(1+\alpha \right)p_{\mathrm{Enc}}^{\left(\epsilon \right)}}{1+\alpha \left(p_{\mathrm{Enc}}^{\left(\epsilon \right)}+\frac{1-p_{\mathrm{Enc}}^{\left(\epsilon \right)}}{K}\right)}>p_{\mathrm{Enc}}^{\left(\epsilon \right)}.$$



**Proof.** Let us consider *K* encoders, each processing features independently. The probability that exactly *s* out of *K* encoders make correct predictions follows a binomial distribution:(20)P(A)=KspEnc(ϵ)s1−pEnc(ϵ)K−s,
where the event set A:={exactlysencoderspredictcorrectly}. If the gate network assigns a sample to any one of these *s* correct encoders, MoEFormer will predict the sample correctly. The conditional probability that MoEFormer predicts correctly, given A, is(21)$$P\left(B|A\right)=\frac{(1+\alpha )s}{\left(1+\alpha )s+(K-s\right)}=\frac{(1+\alpha )s}{K+\alpha s},$$
where B:={MoEFormerpredictscorrectly}. By the law of total probability, the probability of MoEFormer making a correct prediction is(22)P(B)=∑AP(A)P(B∣A)=∑s=1ßKKspEnc(ϵ)s1−pEnc(ϵ)K−s(1+α)sK+αs.Rewriting the summation in terms of t=s−1, we obtain(23)P(B)=∑t=0K−1Kt+1pEnc(ϵ)t+11−pEnc(ϵ)K−1−t(1+α)(t+1)K+α(t+1)=∑t=0K−1K−1tpEnc(ϵ)t1−pEnc(ϵ)K−1−t(1+α)KpEnc(ϵ)K+α(t+1)=E(1+α)KpEnc(ϵ)K+α(t+1),
where the expectation is taken over $t∼\mathrm{Bin}(K-1,p_{\mathrm{Enc}}^{\left(\epsilon \right)})$. We define(24)$$f\left(t\right)=\frac{\left(1+\alpha \right)Kp_{\mathrm{Enc}}^{\left(\epsilon \right)}}{K+\alpha (t+1)}.$$Since f(t) is convex with respect to *t* (as its second derivative is positive), by Jensen’s inequality, we have(25)$$E\left[f\left(t\right)\right]\geq f\left(E[t]\right).$$The expected value of *t* is(26)$$E\left[t\right]=\left(K-1\right)p_{\mathrm{Enc}}^{\left(\epsilon \right)}.$$Substituting this into (Equation 25) yields(27)$$P\left(B\right)\geq f\left(\left(K-1\right)p_{\mathrm{Enc}}^{\left(\epsilon \right)}\right)=\frac{\left(1+\alpha \right)Kp_{\mathrm{Enc}}^{\left(\epsilon \right)}}{K+\alpha \left(\left(K-1\right)p_{\mathrm{Enc}}^{\left(\epsilon \right)}+1\right)}.$$The last expression is strictly increasing with respect to α when α>0, and thus,(28)$$\begin{array}{c} p_{\mathrm{MoE}}^{\left(\epsilon \right)}\geq \frac{\left(1+\alpha \right)p_{\mathrm{Enc}}^{\left(\epsilon \right)}}{1+\alpha \left(p_{\mathrm{Enc}}^{\left(\epsilon \right)}+\frac{1-p_{\mathrm{Enc}}^{\left(\epsilon \right)}}{K}\right)}>\frac{\left(1+0\right)p_{\mathrm{Enc}}^{\left(\epsilon \right)}}{1+0(p_{\mathrm{Enc}}^{\left(\epsilon \right)}+\frac{1-p_{\mathrm{Enc}}^{\left(\epsilon \right)}}{K})}=p_{\mathrm{Enc}}^{\left(\epsilon \right)}. \end{array}$$□

Theorem 1 establishes that the prediction error bound for MoEFormer, $p_{\mathrm{MoE}}^{\left(\epsilon \right)}$, is always better than that of a single encoder model, $p_{\mathrm{Enc}}^{\left(\epsilon \right)}$. Furthermore, the lower bound,(29)$$\frac{\left(1+\alpha \right)p_{\mathrm{Enc}}^{\left(\epsilon \right)}}{1+\alpha \left(p_{\mathrm{Enc}}^{\left(\epsilon \right)}+\frac{1-p_{\mathrm{Enc}}^{\left(\epsilon \right)}}{K}\right)}$$
increases monotonically with α. Intuitively, as α increases (indicating better gate network performance), the lower bound on MoEFormer’s prediction accuracy improves, highlighting the importance of the gate network in leveraging cross-condition knowledge.

## 6. Experiments and Results

### 6.1. Settings

#### 6.1.1. Dataset

The C-MAPSS dataset, developed by NASA using the Commercial Modular Aero-Propulsion System Simulation (C-MAPSS), is a widely recognized benchmark for research in Remaining Useful Life (RUL) prediction for turbofan engines. This dataset simulates the degradation process of turbofan engines under various operating conditions and fault modes, making it highly relevant for evaluating the performance of RUL prediction models. The dataset is divided into four subsets, FD001, FD002, FD003, and FD004, each representing different operational scenarios and fault conditions. These subsets include both training and testing data. The training data contain the full life-cycle degradation information of multiple engines, while the test data comprise incomplete engine life cycles that terminate before failure.

The primary goal is to predict the Remaining Useful Life (RUL) from the sensor readings, which consist of 21 sensor measurements and 3 operational parameters, including temperature, pressure, and speed, collected from various engine components. In this study, we focus on subsets FD002 and FD004 because they include multiple working conditions, aligning with our emphasis on RUL prediction under diverse operational scenarios. These subsets offer a richer dataset for analyzing how variations in engine states and conditions influence degradation dynamics. The training data in these subsets provide complete degradation trajectories, while the test data conclude before failure, simulating real-world scenarios where engines operate under dynamic and changing conditions. By leveraging these subsets, we validate the proposed method for multi-condition RUL estimation, ensuring robustness and generalizability to real-world applications. We follow the widely used data preprocessing step as detailed in [27].

#### 6.1.2. Baselines

To demonstrate the effectiveness of the proposed MoEFormer model for multi-condition RUL prediction, we compare its performance against several state-of-the-art models in the field, including the following:**CNN** [6]: A convolutional-neural-network-based regression approach.**MODBNE** [36]: A multi-objective deep belief network ensemble method.**Dual-Task LSTM** [39]: A dual-task LSTM designed for joint learning of degradation assessment and RUL prediction.**DCNN** [24]: A deep convolutional neural network model that focuses on extracting local data features to improve RUL prediction.**CNN-LSTM** [37]: A hybrid weighted deep domain adaptation approach that combines CNN and LSTM architectures.**Cap-LSTM** [8]: A model that integrates capsule neural networks with LSTM modules for enhanced feature representation.**BiRNN-ED** [40]: An improved similarity-based prognostic model utilizing a bidirectional RNN with an encoder–decoder architecture.**CNN-BiGRU** [27]: A feature-fusion-based method that dynamically adjusts the weights of input features for RUL prediction.**GCU-Transformer** [41]: A Transformer-based architecture incorporating gated convolutional units for improved performance.**HDNN** [42]: A hybrid deep neural network that combines LSTM and CNN layers to extract both temporal and spatial features.

These models represent a wide range of advanced techniques, including convolutional networks, recurrent architectures, hybrid frameworks, and Transformer-based methods. Since most papers do not provide detailed code or hyperparameter configurations, we replicated their settings as described and directly extracted their reported results. For our proposed method, MoEformer, we used the following configuration: a window size of 50, λ=0.7 (the weight of the auxiliary loss), a learning rate of 0.01, and training for 100 epochs with early stopping to prevent overfitting.

#### 6.1.3. Performance Metrics

The evaluation of RUL prediction accuracy is conducted using two widely adopted metrics: the scoring function and the root mean squared error (RMSE).

The scoring function is specifically designed to account for the asymmetry in the consequences of prediction errors. It penalizes delayed predictions (when the predicted RUL exceeds the actual RUL) more heavily than advanced predictions (when the predicted RUL is less than the actual RUL). This asymmetry reflects real-world considerations, where delayed predictions may lead to catastrophic failures, whereas advanced predictions typically result in only minor disruptions due to premature maintenance. The scoring function is defined as follows:$$\mathrm{Score}=\sum_{i=1}^{N}\left\{\begin{array}{cc} \exp\left(\frac{r-\hat{r}}{a_{1}}\right)-1 & \mathrm{if}\;\hat{r}<r,\\ \exp\left(\frac{\hat{r}-r}{a_{2}}\right)-1 & \mathrm{if}\;\hat{r}\geq r, \end{array}\right.$$
where *N* is the total number of samples, *r* is the actual RUL, $\hat{r}$ is the predicted RUL, and $a_{1}=13$, $a_{2}=10$ are the penalty coefficients [27]. By applying heavier penalties to delayed predictions, this metric ensures that models are incentivized to prioritize safety-critical predictions.

RMSE is a more traditional evaluation metric that measures the overall magnitude of prediction errors without differentiating between advanced and delayed predictions. RMSE is defined as$$\mathrm{RMSE}=\sqrt{\frac{1}{N}\sum_{i=1}^{N}{\left(\hat{r}-r\right)}^{2}}.$$

RMSE provides an aggregate measure of prediction accuracy, with lower values indicating better performance. While it effectively quantifies the overall error magnitude, RMSE does not capture the asymmetric nature of prediction errors. As a result, it is less informative in scenarios where the timing of maintenance actions is critical.

### 6.2. Results

Table 1 summarizes the predictive performance of 11 methods across the four sub-datasets of the C-MAPSS dataset. For the single-condition datasets, FD001 and FD003, all methods achieved relatively strong performance, with minor differences in predictive accuracy. Although MoEFormer did not outperform all methods on these datasets, its performance was comparable to the best-performing approaches. In contrast, the multi-condition datasets, FD002 and FD004, presented greater challenges, as evidenced by the more pronounced performance gaps among the methods. On these more complex datasets, the proposed MoEFormer consistently delivered lower prediction errors than competing approaches. Notably, in terms of the Score metric, MoEFormer demonstrated a significant advantage over the second-best method, achieving reductions of 38.2% and 35% on FD002 and FD004, respectively. These results highlight MoEFormer’s superior capability in addressing the complexities of multi-condition RUL prediction tasks.

Figure 2 plots the real and predicted RUL for FD002 and FD004 of individual test samples (i.e., test engines). The x-axis corresponds to the index of test samples. Following the visualization method in [8], we connect the points into a curve to better illustrate the differences between the real and predicted RUL values. The results indicate that MoEFormer achieves high accuracy in most cases across these multi-condition datasets. However, for instances where the actual RUL is significantly large, MoEFormer tends to slightly underestimate the RUL. While this introduces a minor prediction bias, such early-warning errors are generally more acceptable than delayed predictions, as they allow for proactive maintenance scheduling. This aligns with the advantages reflected in the Score metric (Table 1), where MoEFormer demonstrates its robust performance.

Figure 3 further illustrates the sample-wise distribution of MoEFormer’s predicted RUL values against the actual RUL values for FD002 and FD004. Most data points cluster near the anti-diagonal, indicating a strong agreement between predictions and ground truth. Notably, predictions for shorter actual RUL values (points near the bottom-left corner) align closely with the ideal diagonal, whereas predictions for longer RUL values (points near the top-right corner) exhibit slightly higher variance. This observation highlights the inherent difficulty of predicting longer RUL values. Interestingly, a horizontal line appears at actual RUL = 125, reflecting a practical constraint in the dataset where engines are retired once their cycles reach 125, irrespective of their true condition. This operational cutoff causes predictions in this region to slightly overestimate the actual values. Despite this artificial intervention, the prediction errors remain well within acceptable bounds, further demonstrating MoEFormer’s reliability and effectiveness in tackling multi-condition RUL prediction tasks.

### 6.3. Ablation Study

This subsection presents an ablation study on the two key components of MoEFormer: distribution alignment (DA) and Mixture of Encoders (MoE). The study is conducted on the FD002 and FD004 datasets to evaluate the individual and combined contributions of these components. Specifically, three ablated versions of the model are analyzed: “w/o DA”, “w/o MoE”, and “w/o DA&MoE”, where DA, MoE, or both components are progressively removed. Table 2 compares the performance of these ablated models with the full MoEFormer.

The results reveal that removing both DA and MoE (“w/o DA&MoE”) leads to a significant deterioration in performance, characterized by large prediction errors and high Score values. As DA and MoE are gradually reintroduced, the prediction errors steadily decrease, demonstrating the critical importance of both components. Interestingly, DA and MoE exhibit distinct yet complementary roles across the two datasets: DA has a greater impact on FD002, while MoE contributes more significantly to the performance on FD004. These findings underscore the necessity of combining both components to achieve robust and accurate multi-condition RUL predictions. The complete MoEFormer model consistently delivers the best performance, validating the design of its key components.

### 6.4. Study on Hyperparameters

This section evaluates the effect of the two key hyperparameters, the window length *L* and the auxiliary loss weight λ, on MoEFormer’s prediction performance. Figure 4a illustrates the variation in RMSE as a function of *L* on the FD002 and FD004 datasets. A similar trend is observed for both datasets: RMSE decreases initially, stabilizes within the range of L=40 to 60, and then increases when *L* becomes excessively large. The best performance is achieved at L=50.

In contrast, as shown in Figure 4b, the Score metric follows a similar overall trend but reveals some dataset-specific differences. When L>60, the Score on FD002 increases sharply, while it remains relatively stable on FD004. This discrepancy arises from the Score metric’s sensitivity to late-prediction errors. Specifically, as *L* increases, MoEFormer tends to overpredict RUL values on FD002 but slightly underpredicts them on FD004. This behavior likely reflects differences in the underlying data distributions of the two datasets. Overall, these results indicate that while MoEFormer’s performance is influenced by the choice of window length *L*, the model remains robust within a reasonable range. Optimal values for *L* can be easily determined through simple validation experiments, with L=50 being a reliable choice for both datasets.

The performance of MoEFormer as a function of λ is shown in Figure 5. The results show that λ>0 outperforms λ=0, demonstrating the auxiliary loss’s effectiveness. However, the performance does not change monotonically with λ>0, displaying irregular fluctuations depending on the metric. For instance, λ=0.3 achieved the best RMSE on FD004, whereas λ=0.7 provided the optimal Score on FD002, reflecting the metrics’ differing emphases: MSE penalizes overall error, while Score penalizes delayed predictions. Overall, λ=0.7 provided the best trade-off across datasets and metrics. As a rough suggestion, it is recommended to set λ to a value between 0.2∼1.0, but the best choice might need adjustment for different scenarios through cross validation.

## 7. Conclusions

In this paper, we proposed MoEFormer, a novel framework for accurate multi-condition Remaining Useful Life (RUL) prediction. By combining a Mixture of Encoders (MoE) with a Transformer-based architecture, MoEFormer addresses the critical challenge of leveraging cross-condition knowledge while maintaining condition-specific feature extraction. The dynamic gated mixture module enables the model to seamlessly integrate shared and condition-specific knowledge, while the Transformer component effectively captures temporal dependencies. The proposed approach is further supported by a theoretical lower bound on the error rate, offering a formal performance guarantee. Extensive experiments conducted on the C-MAPSS dataset validate the effectiveness of MoEFormer, showing significant improvements in predictive accuracy and robustness over state-of-the-art methods, including a 38.2% and 35% reduction in the Score metric on the FD002 and FD004 datasets, respectively. This work not only advances the state of the art in RUL prediction but also highlights the importance of dynamic and adaptive architectures for handling diverse operational conditions. The integration of condition-specific expertise and cross-condition learning represents a meaningful step forward in the domains of prognostics and health management.

Despite the promising results, there are several directions for future research. First, while MoEFormer effectively models multi-condition scenarios, further exploration is needed to adapt the framework for scenarios with limited labeled data through techniques such as transfer learning or semi-supervised learning. Second, incorporating uncertainty quantification into the predictions could improve the reliability of the model for real-world applications, particularly in safety-critical industries. Third, expanding the application of MoEFormer to other datasets and domains, such as renewable energy systems or autonomous vehicles, could validate its versatility and generalization capabilities. Lastly, optimizing the computational efficiency of the model, especially for real-time deployments, remains an important area for future improvement.

## Figures and Tables

**Figure 1 entropy-27-00079-f001:**
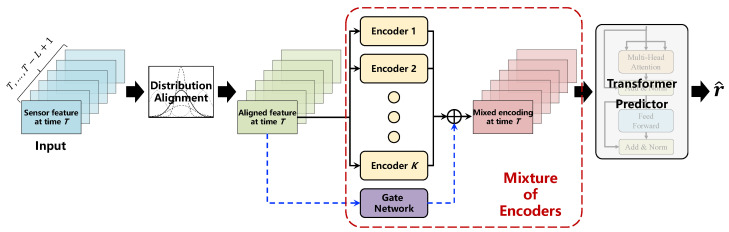
Framework of the proposed MoEFormer. The model comprises three key components: distribution alignment, Mixture of Encoders (MoE), and Transformer predictor. The distribution alignment step aims to mitigate distribution drift in sensor features collected under varying operating conditions. The MoE component is designed to extract complementary knowledge from different working conditions. Lastly, the Transformer predictor captures temporal dependencies within the features, ultimately producing the final RUL prediction.

**Figure 2 entropy-27-00079-f002:**
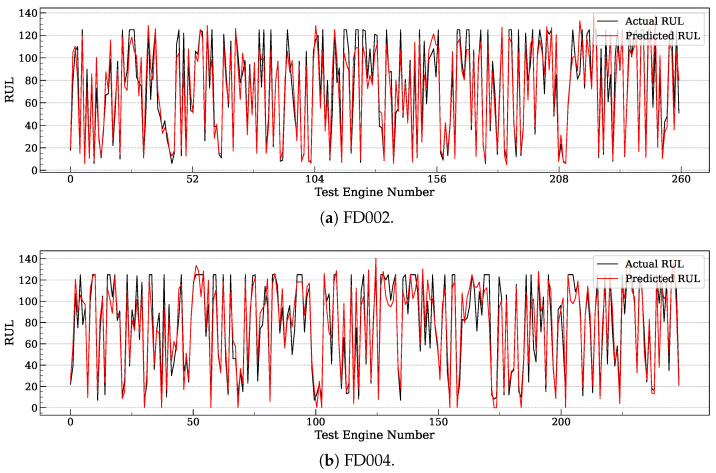
Comparison of MoEFormer’s predicted RUL values against the ground truth for FD002 and FD004.

**Figure 3 entropy-27-00079-f003:**
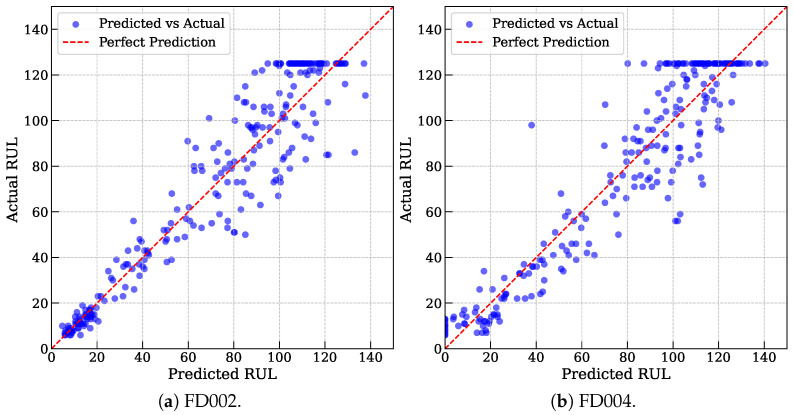
The sample-wise distribution of MoEFormer’s predicted RUL values against the actual RUL values for FD002 and FD004.

**Figure 4 entropy-27-00079-f004:**
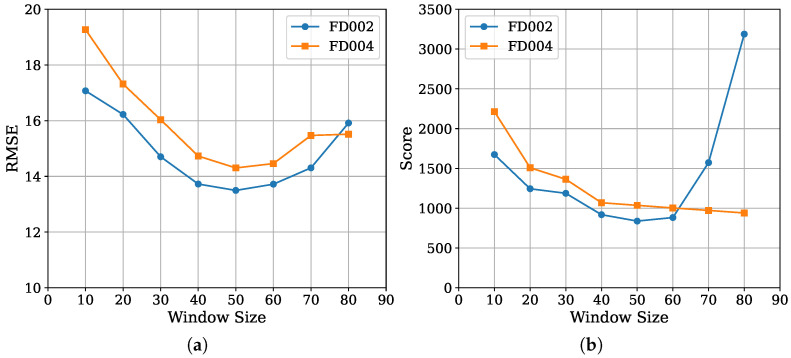
(**a**) RMSE and (**b**) Score of MoEFormer as a function of window size.

**Figure 5 entropy-27-00079-f005:**
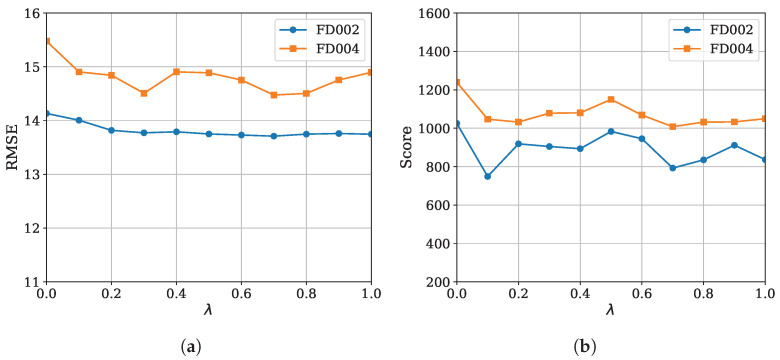
(**a**) RMSE and (**b**) Score of MoEFormer as a function of λ (refer to (Equation 17)).

**Table 1 entropy-27-00079-t001:** Performance comparison on C-MAPSS sub-datasets. The best performance is reported using **bold**, and the second best is reported using an underline. “-” means the results are not released. Metrics with a downward arrow ↓ indicate smaller values and better performance.

Model	FD001		FD002		FD003		FD004	
	**(Single-Condition)**		**(Multi-Condition)**		**(Single-Condition)**		**(Multi-Condition)**	
	**RMSE**	**Scorex↓**	**RMSE**	**Score↓**	**RMSE**	**Score↓**	**RMSE**	**Score↓**
CNN	18.45	1286.7	30.29	13,570	19.82	1596.2	29.16	7886.4
MODBNE	15.04	334.2	25.05	5585.3	12.51	421.9	28.66	6557.6
Dual-Task LSTM	12.29	-	17.87	-	14.34	-	21.81	-
DCNN	12.61	273.7	22.36	10,412	12.64	284.1	23.31	12,466
CNN-LSTM	14.40	290.0	27.36	9869.0	14.32	316.0	26.69	6594.0
Cap-LSTM	12.27	260.0	17.19	1850.0	12.55	**217.0**	22.05	4570.0
BiRNN-ED	13.58	**228.0**	19.59	2650.0	19.16	1727.0	22.15	2901.0
CNN-BiGRU	12.31	252.1	16.06	1238.1	12.37	283.5	19.83	2706.8
GCU-Transformer	**11.27**	-	22.81	-	**11.42**	-	24.86	-
HDNN	13.02	245.0	15.24	1282.4	12.22	287.7	18.16	1527.4
MoEFormer	15.13	395.60	**13.71**	**792.64**	13.88	290.67	**14.47**	**1008.25**

**Table 2 entropy-27-00079-t002:** Ablation study on two key designs of MoEFormer: distribution alignment (DA) and mixture of encoders (MoE). The best performance is reported using **bold**, and the second best is reported using an underline. Metrics with a downward arrow ↓ indicate smaller values and better performance.

Model	FD002		FD004	
	**RMSE**	**Score↓**	**RMSE**	**Score↓**
MoEFormer w/o DA&MoE	30.01	34,464.83	33.99	89,148.18
MoEFormer w/o DA	20.76	2450.17	28.53	10,390.44
MoEFormer w/o MoE	24.08	6267.54	23.60	4763.35
MoEFormer	**13.71**	**792.64**	**14.47**	**1008.25**

## Data Availability

The original data presented in the study are openly available in NASA’s Open Data Portal at https://data.nasa.gov/Aerospace/CMAPSS-Jet-Engine-Simulated-Data/ff5v-kuh6/about_data, accessed on 13 January 2025.

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
