# Peer review of "Multi-Condition Remaining Useful Life Prediction Based on Mixture of Encoders"

_entropy, 2025, doi:10.3390/e27010079_

Round 1
Reviewer 1 Report
Comments and Suggestions for Authors
For varying operational conditions, this paper introduces MoEFormer, which combines a Mixture of Encoders (MoE) with a Transformer-based architecture to achieve precise multi-condition RUL prediction. The topic is interesting, and the experimental results seems that MoEFormer outperforms several state-of-the-art methods for multi-condition RUL prediction. Some comments are as follows.
1. In section 4.1, the authors presents a distribution alignment method. But we know that in the operation of equipment, it is not only different working conditions, but also different health states at different times. In this case, is such alignment still meaningful?
2. In formula (17), λ = 0.7 in this study is used. How should this parameter be selected, is it only suitable for the example in this article, or is there any guidance?
3. In Theorem 1, the authors claim that suppose K encoders process features independently, but how to guarantee this condition?
4. In table1, what are the neural network structures and parameters used for each method? This also has a significant impact on the results.
5. In figure 2, what is the vertical axis? The remaining life curve should be a monotonically decreasing curve.
Reviewer 2 Report
Comments and Suggestions for Authors
This paper addresses the task of Remaining Useful Life (RUL) prediction and identifies the limitation of previous methods in effectively leveraging cross-condition knowledge for multi-condition RUL prediction tasks. These methods typically rely on a single model that handles both single-condition and multi-condition RUL predictions separately, without sharing knowledge across conditions. To address this issue, the authors propose a novel model for multi-condition RUL prediction, named MoEFormer, which combines a Mixture of Experts (MoE) encoder with a Transformer-based architecture to achieve more accurate multi-condition RUL predictions.
Specifically, MoEFormer utilizes multiple expert encoders, each dedicated to extracting features for a specific operational condition, enabling the model to capture condition-specific knowledge effectively. Additionally, the Transformer architecture is employed to capture long-term temporal dependencies, further enhancing the model's ability to model sequential information.
The paper demonstrates the effectiveness of the proposed framework both theoretically and experimentally. Theoretically, the authors derive a lower bound for the multi-condition error rate, establishing the feasibility of the approach. Experimentally, extensive tests on the widely-used C-MAPSS dataset show that MoEFormer outperforms state-of-the-art methods under multi-condition scenarios.
Questions
1. Although the model performs well under multi-condition scenarios, its performance is relatively modest under single-condition settings. Could you provide an explanation for this discrepancy?
2. Are there any previous methods addressing multi-condition RUL prediction in the literature? If so, it would be valuable to discuss how your approach compares to these methods and highlight the advances of your model.
3. The theoretical analysis in Section 5 is interesting, but the meaning of the last bound (Equation 29) does not seem intuitive. Could you explain this in more detail?
Round 2
Reviewer 1 Report
Comments and Suggestions for Authors
All my concerns have been addressed, I recommand to accept the paper.